

# Modeling using clinical examination indicators predicts interstitial lung disease among patients with rheumatoid arthritis

Yao Wang[1,2], Wuqi Song[1], Jing Wu[1], Zhangming Li[3], Fengyun Mu[4], Yang Li[5], He Huang[5], Wenliang Zhu[6] and Fengmin Zhang[1]

[1] Department of Microbiology, Wu Lien-Teh Institute, Harbin Medical University, Harbin, Heilongjiang Province, China
[2] Department of Microbiology and Immunology, School of Basic Medical Sciences, Heilongjiang University of Chinese Medicine, Harbin, Heilongjiang Province, China
[3] Department of Pharmacy Administration, Harbin Medical University, Harbin, Heilongjiang Province, China
[4] Department of Laboratory Medicine, The Second Affiliated Hospital of Harbin Medical University, Harbin, Heilongjiang Province, China
[5] Department of Rheumatology, The Second Affiliated Hospital of Harbin Medical University, Harbin, Heilongjiang Province, China
[6] Institute of Clinical Pharmacology, The Second Affiliated Hospital of Harbin Medical University, Harbin, Heilongjiang Province, China

## ABSTRACT

Interstitial lung disease (ILD) is a severe extra-articular manifestation of rheumatoid arthritis (RA) that is well-defined as a chronic systemic autoimmune disease. A proportion of patients with RA-associated ILD (RA-ILD) develop pulmonary fibrosis (PF), resulting in poor prognosis and increased lifetime risk. We investigated whether routine clinical examination indicators (CEIs) could be used to identify RA patients with high PF risk. A total of 533 patients with established RA were recruited in this study for model building and 32 CEIs were measured for each of them. To identify PF risk, a new artificial neural network (ANN) was built, in which inputs were generated by calculating Euclidean distance of CEIs between patients. Receiver operating characteristic curve analysis indicated that the ANN performed well in predicting the PF risk (Youden index = 0.436) by only incorporating four CEIs including age, eosinophil count, platelet count, and white blood cell count. A set of 218 RA patients with healthy lungs or suffering from ILD and a set of 87 RA patients suffering from PF were used for independent validation. Results showed that the model successfully identified ILD and PF with a true positive rate of 84.9% and 82.8%, respectively. The present study suggests that model integration of multiple routine CEIs contributes to identification of potential PF risk among patients with RA.

Corresponding author
Fengmin Zhang,
fengminzhang@ems.hrbmu.edu.cn

## INTRODUCTION

Rheumatoid arthritis (RA) is a common chronic systemic autoimmune disorder mainly characterized by joint inflammation. Apart from articular tissue, multiple other tissues and organs may be involved in the pathological process of RA. Indeed, extra-articular manifestations (EAMs) have become the main cause of the morbidity and mortality of patients with RA (*Turesson, 2013*; *Sihvonen et al., 2004*). Among the recognized EAMs, interstitial lung disease (ILD) follows cardiac manifestations (*Brown, 2007*) as the second contributor to the excess mortality (10%–20%) of patients with RA (*Ingegnoli et al., 2012*). Compared with the median survival of 9.9 years for RA alone, patients with RA-associated ILD (RA-ILD) have been reported to have poor prognosis with a median survival of 2.6 years (*Bongartz et al., 2010*). However, the situation might be worse if pulmonary fibrosis (PF) is also confirmed. In a clinical study focusing on fibrotic interstitial pneumonia, Solomon and colleagues (*2013*) verified that RA patients with fibrotic ILD had worse survival than those with non-fibrotic ILD, indicating that PF is an independent risk for mortality in RA.

PF causes the aggressive deterioration of lung function and leads to poor prognosis of RA. Unfortunately, fibrotic ILD, especially the subtype usual interstitial pneumonia, still lacks targeted therapy (*Travis et al., 2008*). This leads to worse outcomes in such patients. Nevertheless, early identification of patients with high PF risk would definitely benefit individual RA patient management, in which joint participation of multidisciplinary doctors has been suggested for driving treatment decisions (*Lake & Proudman, 2014*). In the clinical context, any decision making is dependent on diversified clinical examinations including a large number of clinical examination indicators (CEIs). The collection of numerous CEIs comprehensively reflects the current pathophysiological condition of the patient. In this study, we hypothesized that integration of the CEIs may reveal the potential risk of an individual patient suffering from PF. If the assumption is valid, early risk assessment may be made on admission. To validate the rationality and feasibility of this hypothesis, we performed a retrospective study, in which 620 patients with established RA were included and their clinical examination results were retrieved from the electronic medical records system of the hospital. A novel artificial neural network (ANN) was considered for abstracting and integrating significant information related to PF risk from CEIs. Especially, rather than the traditional ANN and neural network cascade previously described (*Zhu & Kan, 2014*; *Li et al., 2015*; *Hou et al., 2016*), the new ANN built here was thought to be a derivative information integration system, in which inputs were generated by calculating the Euclidean distances of CEIs between patients. In conclusion, such an effort aims to provide a viable clinical approach to identify patients with high PF risk and facilitate implementation of early preventive interventions.

## METHODS

### Ethical statement

This study is a retrospective study that was approved by the Ethics Committee of Harbin Medical University (HMU) (Approval number: HMUIRB20150028) and was carried out in accordance with the Declaration of Helsinki.

## Patients

The electronic medical record systems of the first and second affiliated hospitals of HMU were used to access the medical records of hospitalized patients that were clinically diagnosed with RA For each patient, high-resolution computed tomography (HRCT, Discovery CT 750 HD; GE Medical Systems, LLC., Waukesha, WI, USA) was performed to examine whether the complication of ILD or PF was present. In the second affiliated hospital of HMU, 32 CEIs were retrieved from the hospital's electronic system of medical records (form January 1, 2013 to December 31, 2015). All the CEIs were categorized into three classes. The three classes were basic information (two items), routine blood test (22 items), and routine urine test (eight items). In the first affiliated hospital of HMU, four CEIs including age, eosinophil count (EO), platelet count (PLT), and white blood cell count (WBC) were retrieved from the hospital's electronic system of medical records (from January 1, 2014 to December 31, 2015). If a patient's clinical examination was inadequate, he or she was not taken into consideration for inclusion as a subject. It should be especially noted that there were no other reasons for rejecting subject inclusion, such as age or gender.

## Identification of ILD-associated CEIs

The software MedCalc v15.8 (MedCalc, Mariakerke, Belgium) was used to perform receiver operating characteristic (ROC) curve analysis and calculate the Youden index for each CEI. The Youden index was the sum of sensitivity and specificity minus 1 as defined previously (*Youden, 1950*). This effort aimed to investigate whether a CEI can be used as a marker to distinguish patients suffering from ILD from those with healthy lungs. It should be noted that only the RA patients with healthy lungs and those suffering from ILD were included in the calculation of the Youden index and the subsequent construction of networks and models. ILD-associated CEIs were identified only when the area under the ROC curve (AUC) was significantly larger than 0.5 ($P < 0.05$), and were retained for further integration by ANN.

## Data preprocessing and model integration of ILD-associated CEIs

ILD-associated CEIs were normalized into a 0–1 number before further analysis as previously described (*Zhu & Kan, 2014*). The Intelligent Problem Solver (IPS) tool in the software STATISTICA Neural Networks (SNN, Release 4.0E; Statsoft, Tulsa, OK, USA) was applied to construct a radial basis function (RBF)-ANN model to investigate the effect of CEI integration on ILD association. In this study, the model was simply named as ANN I. The model output then underwent normalization processing and Youden index calculation to investigate whether an obvious association with ILD status was still present after CEI integration. The holdout cross-validation method was applied for preliminary validation of the model as IPS randomly divided the patients into three subsets (training set, verification set, and testing set) in a 2:1:1 ratio. Thus, one-quarter of all the patients did not participate in model building and were used for model testing. The IPS calculated correlation coefficients for the training set ($R_{Tr}$) and the testing set ($R_{Te}$). The two correlation coefficients measured the correlation between model output and status of ILD. Similar values of $R_{Tr}$ and $R_{Te}$ indicates good generalization ability of the model.
### Euclidean distance calculation and construction of patient–patient similarity network

For any two patients, we calculated their Euclidean distance in an $n$-dimensional space, in cases in which ILD-associated CEIs were re-defined as space coordinates. The value of $n$ was the sum of ILD-associated CEIs. Following the clustering algorithm proposed by *Rodriguez & Laio (2014)*, we established a patient–patient similarity network (PPSN) by using the network data visualization software Cytoscape v2.8.3 (Institute of Systems Biology, Seattle, WA, USA) (*Smoot et al., 2011*).

### Model integration of derivative information of patients

Construction of the ANN I model and the calculation of Euclidean distance of CEIs between patients was used to obtain derivative information for each patient. For example, four items of derivative information could be obtained for patient $i$ as follows: first, we divided the other patients except patient $i$ into $m$ mutually exclusive divisions of nearly equal size according to the magnitude of their distances to patient $i$ in the $n$-dimensional space of ILD-associated CEIs. Thus, the four items of derivative information might be the ANN I output of patient $i$, mean ANN I output of patients in a given division, mean Euclidean distance to the patients in the division, and actual proportion of RA-ILD among patients in the division. The four derivative information items were imported in a new RBF-ANN as inputs to predict patients' potential PF risk, in the same way as the ANN I. In order to distinguish it from ANN I, the new ANN was named as ANN II. We further investigated the effect of different patient groupings on the performance of ANN II. In this study, the division size $m$ was assigned a value of 5, 10, 15 or 20. For example, if we divided patients into five divisions, we could obtain five candidates of ANN II. The Youden index calculation was then used to identify the best model as ANN II.

### Model validation and performance evaluation

For ANN II, the 10-fold cross-validation method was used for model validation as previously described (*Li et al., 2015*). Briefly, all the patients were randomly divided into 10 mutually-exclusive sets of nearly equal size. Next, nine were selected for model training and one was used for model validation. The above procedure was repeated 10 times to allow each of the 10 patient sets to be independently used for validation.

To investigate whether the ANN models I and II identify patients with high PF risk, we performed ROC curve analysis on ILD-associated CEIs and outputs of ANN I and ANN II using MedCalc v15.8. Besides AUC we also recorded the values of sensitivity, specificity, Youden index, and calculated diagnostic odds ratio (DOR) at the optimal cut-off point. The Youden index was the sum of sensitivity and specificity minus 1 as defined previously (*Youden, 1950*). According to the definition of DOR in previous studies (*Böhning, Holling & Patilea, 2011*; *Glas et al., 2003*), it was calculated as follows:

$$\text{DOR} = \frac{\text{sensitivity} \times \text{specificity}}{(1 - \text{sensitivity}) \times (1 - \text{specificity})}.$$

In addition, a set of 87 RA patients suffering from PF (the second affiliated hospital of HMU) was used for independent evaluation of each ILD-associated CEI and ANN models

I and II. A further set consisting of 72 RA patients with healthy lungs and 146 RA patients suffering ILD (the first affiliated hospital of HMU) was used for external validation of ANN model II. The true positive rate (TPR) was calculated when the optimal cut-off point in the ROC curve was used as the discriminant threshold. TPR was calculated as follows:

$$TPR = \frac{TP}{TP + FN} \times 100\%$$

where TP and FN are abbreviations of true positive and false negative.

## Statistical analysis

MedCalc v15.8 was used to perform pairwise comparisons of the ROC curves based on the methodology of *DeLong, DeLong & Clarke-Pearson (1988)*. Differences were considered as statistically significant when $P < 0.05$.

## RESULTS

### Patients

A total of 838 patients with RA were included in this study. Among them, 620 patients from the second affiliated hospital of HMU were subjected to complete clinical examination (Tables S1 and S2). HRCT examination verified the complication of PF in 87 of the 620 patients. In addition, 169 patients were identified as having healthy lungs and 364 were diagnosed as complicated with ILD. For 218 patients with RA from the first affiliated hospital of HMU, only four CEIs were retrieved from the hospital's electronic system of medical records (Table S3), among which 72 were identified as having healthy lungs and 146 were diagnosed as complicated with ILD.

### Identification of ILD-associated CEIs

For each of the 32 CEIs (Table S1), we investigated any potential association with the status of ILD. Compared with patients with healthy lungs, those RA patients suffering from ILD have an implied greater PF risk (*Brown, 2007*; *Ingegnoli et al., 2012*). Eight CEIs were identified as ILD-associated CEIs (Table 1). For an ILD-associated CEI, higher Youden index suggested better effectiveness as diagnostic marker of PF risk (Table 1). For instance, among the 32 CEIs, age was assigned the highest Youden index or 0.301 (Fig. 1A). This characteristic was highlighted in the pairwise comparisons of the ROC curves as compared with EO, PLT or WBC (the methodology of DeLong et al., $P < 0.001$, Fig. 1B). The Youden index calculation indicated that age and three blood CEIs (EO, PLT, and WBC) were assigned the highest Youden indices, implying relatively closer association with ILD status (Fig. 1A). Consistent with this, the optimal ANN model effect was obtained by using the four CEIs (age, EO, PLT, and WBC) as joint inputs of ANN I (Fig. S1).

### Euclidean distance calculation for networking patients' similarity

By refining the four ILD-associated CEIs (age, EO, PLT, and WBC) as coordinates in a four-dimensional space, we calculated the Euclidean distance between any two patients and mapped a PPSN for the 533 RA patients including 169 with healthy lungs and 364 complicated with ILD (Fig. 2). In spite of the application of different edge settings, a huge patient cluster, rather than multiple scattered patient clusters, was always observed in the network.

**Table 1** Comparison of ILD-associated CEIs in identifying patients with high PF risk.

| CEI | AUC | Youden index | SE | SP | SE$_{SP=0.8}$ | DOR |
|---|---|---|---|---|---|---|
| Age | 0.710 | 0.301 | 0.74 | 0.56 | 0.47 | 3.63 |
| EO | 0.562 | 0.164 | 0.64 | 0.53 | 0.25 | 1.96 |
| PLT | 0.569 | 0.165 | 0.55 | 0.62 | 0.26 | 1.95 |
| WBC | 0.587 | 0.173 | 0.45 | 0.73 | 0.32 | 2.15 |
| NEUT | 0.577 | 0.133 | 0.85 | 0.28 | .26 | 2.26 |
| U-SG | 0.580 | 0.135 | 0.93 | 0.20 | 0.30 | 3.57 |
| U-WBC | 0.562 | 0.138 | 0.78 | 0.36 | 0.28 | 1.97 |
| U-WBCH | 0.561 | 0.123 | 0.78 | 0.34 | 0.28 | 1.85 |

**Notes.**

CEI, clinical examination indicator; AUC, area under the ROC curve; SE, sensitivity; SP, specificity; SE$_{SP=0.8}$, sensitivity at fixed specificity = 0.8; DOR, diagnostic odd ratio; EO, eosinophil count; PLT, blood platelet count; WBC, white blood cell count; NEUT, neutrophil count; U-SG, urine specific gravity; U-WBC, white blood cell count in urine; U-WBCH, white blood cell (high power field) in urine.

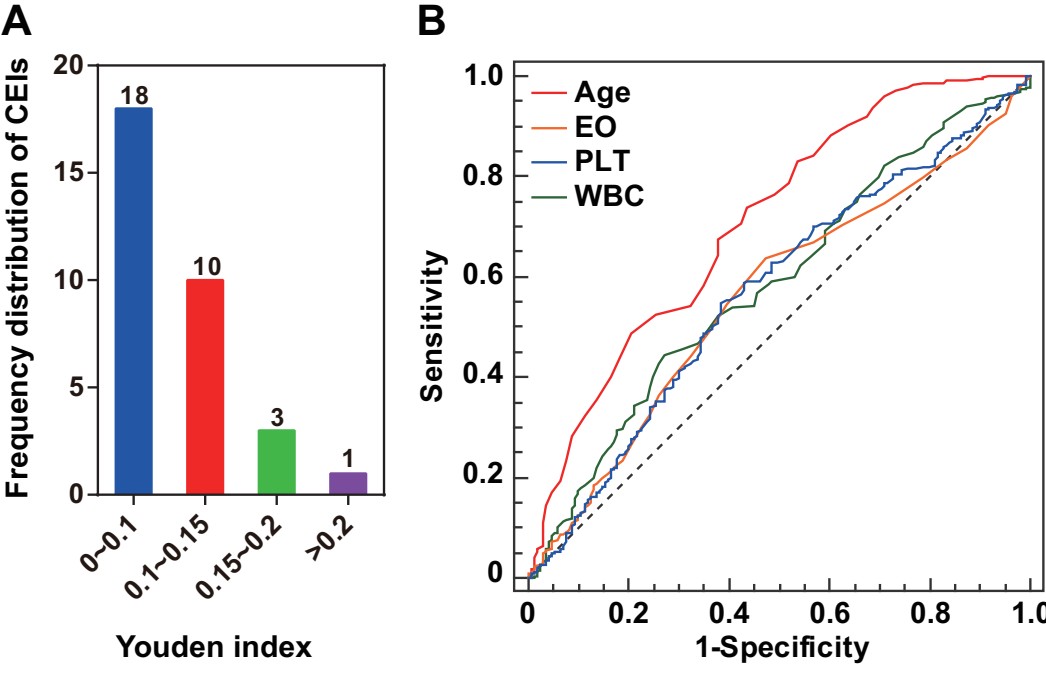

**Figure 1** **ILD-associated CEIs.** (A) Distribution of the 32 CEIs in Youden index value. The numbers on the columns indicate the number of CEIs. Four CEIs were observed to have a Youden index of more than 0.15. (B) ROC curves of the four ILD-associated CEIs age, eosinophil count (EO), blood platelet count (PLT), and white blood cell count (WBC).

## Application of ANN for integration of ILD-associated CEIs

An RBF-ANN with 4-12-1 architecture, named ANN I, was constructed for integration of the four CEIs that were related to PF status, namely age, EO, PLT, and WBC. Compared with single CEIs, the outputs of ANN I had a better ability to identify patients with high PF risk (Youden index = 0.387, Table 2). Furthermore, we built a series of RBF-ANNs with 4-12-1 architecture by calculating Euclidean distance among patients, distributing patients
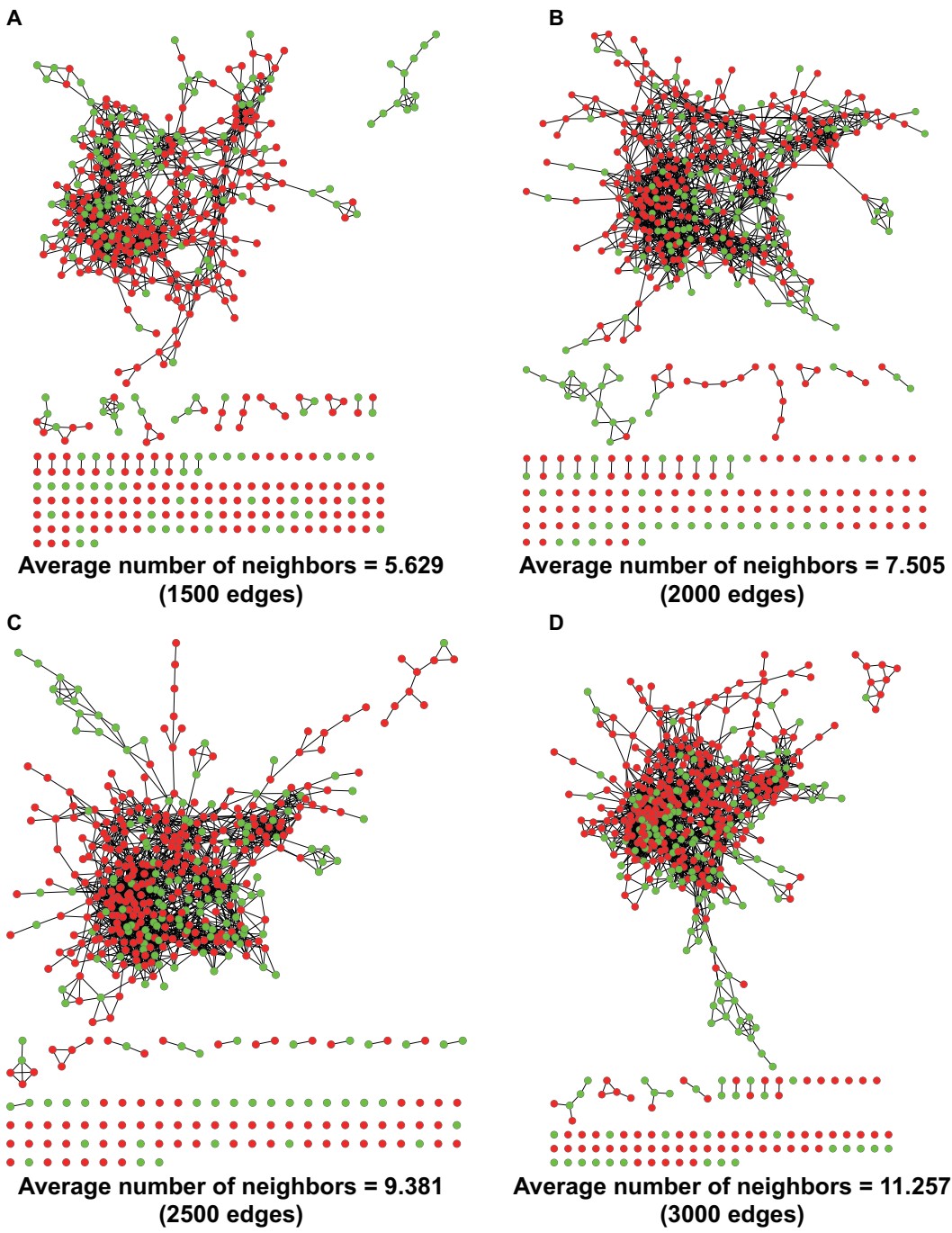

**A**

**Average number of neighbors = 5.629
(1500 edges)**

**B**

**Average number of neighbors = 7.505
(2000 edges)**

**C**

**Average number of neighbors = 9.381
(2500 edges)**

**D**

**Average number of neighbors = 11.257
(3000 edges)**

**Figure 2  PPSNs of different edges.** (A) PPSN of 1,500 edges; (B) PPSN of 2,000 edges; (C) PPSN of 2,500 edges; (D) PPSN of 3,000 edges. Green dots represent patients with healthy lungs and red dots indicate patients complicated with ILD. Edge between two dots that represent patients means a short Euclidean distance between the two patients in a 4-dimensional space, in which the four ILD-associated CEIs were re-defined as coordinates. Although 533 patients were included in the calculation of Euclidean distance, only the shortest distances could be visualized as edges in PPSN. For each edge setting, the average number of neighbors was also calculated.

**Table 2  Comparison of models in identifying patients with high PF risk.**

| Model | AUC | Youden index | SE | SP | SE$_{SP=0.8}$ | DOR |
|-------|-----|--------------|-----|-----|---------------|-----|
| ANN I | 0.734 | 0.387 | 0.772 | 0.615 | 0.473 | 5.41 |
| 5D1 | 0.767 | 0.436 | 0.791 | 0.645 | 0.591 | 6.88 |
| 10D1 | 0.758 | 0.411 | 0.731 | 0.681 | 0.563 | 5.80 |
| 15D1 | 0.746 | 0.407 | 0.780 | 0.627 | 0.489 | 5.96 |
| 20D1 | 0.736 | 0.403 | 0.805 | 0.598 | 0.500 | 6.14 |

**Notes.**

CEI, clinical examination indicator; AUC, area under the ROC curve; SE, sensitivity; SP, specificity; SE$_{SP=0.8}$, sensitivity at fixed specificity $= 0.8$; DOR, diagnostic odd ratio.

5D1, 10D1, 15D1, and 20D1 were the first ANN models created by dividing all the patients with healthy lungs and those complicated with ILD into 5, 10, 15, or 20 groups, respectively.

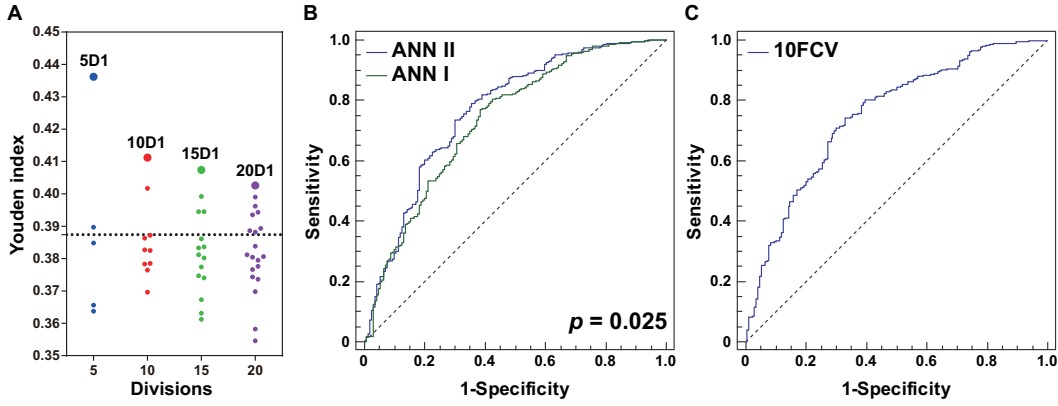

**Figure 3  Optimization and evaluation of ANN II.** (A) Distribution of the Youden index values of ANN models built by dividing all the patients with healthy lungs and those complicated with ILD into 5, 10, 15, or 20 groups, respectively. The first ANN models of different division sizes 5D1, 10D1, 15D1, and 20D1 are highlighted as larger dots. The position of the dotted line is 0.387, and values of Youden index were calculated using ANN I. (B) Comparison of ANNs I and II in ROC curves. A significant difference in AUCs was found ($P = 0.025$). (C) The 10-fold cross-validation (10FCV) result of ANN II.

into divisions, and generating derivative indicators (see 'Methods'). Regardless of the size of the divisions, the ANN generated by the division containing most similar patients showed the best effect in identifying potential PF risks (Fig. 3A). Compared with smaller grouping size, the five division method led to creation of an optimal ANN model, namely ANN II (Youden index $= 0.436$, Table 2). A significant difference was observed in AUC when ANNs I and II were compared ($P = 0.025$, Fig. 3B), suggesting the advantage of ANN II in identifying potential PF risk. The DOR of ANN I was 5.41 and that of ANN II was 6.88. When specificity was fixed as 0.80, the sensitivity of ANN I was 0.473 while that of ANN II was 0.591. These assessments indicated that ANN II was more effective and sensitive than ANN I in identifying patients with high PF risk.

## Evaluation of the models

The holdout cross-validation method was used for preliminary validation of the ANNs established in this study. Two model generalization ability parameters $R_{Tr}$ and $R_{Te}$ were calculated using SNN software. Similar values of $R_{Tr}$ and $R_{Te}$ indicated that the model built

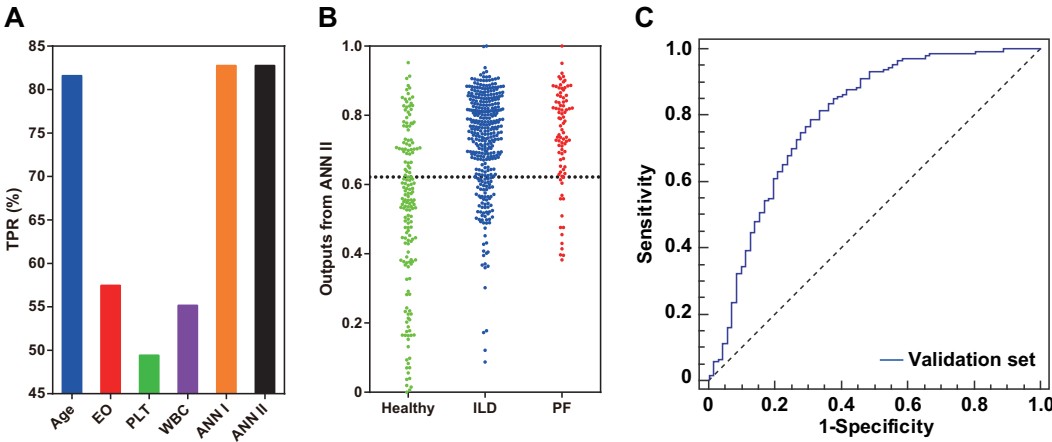

**Figure 4  Model evaluation using an independent patient set of 87 RA patients with PF and an external validation set of 218 RA patients.** (A) Comparison of single ILD-associated CEIs, ANNs I and II in identifying patients with PF. (B) Scatter plots of ANN II outputs for patients with healthy lungs and those complicated with ILD or PF. The position of the dotted line is 0.622, the optimal ROC curve cut-off point of ANN II. (C) The external validation result of ANN II.

using the IPS tool had good generalization ability. For ANN I, $R_{Tr}$ and $R_{Te}$ were 0.398 and 0.440, respectively, while for ANN II, $R_{Tr}$ and $R_{Te}$ were 0.461 and 0.489, respectively. For ANN II in particular, the 10-fold cross-validation method was applied for further model validation (Fig. 3C). The AUC was 0.751, implying effectiveness of ANN II in identifying potential PF risk among the patients. Furthermore, an independent set of 87 RA patients complicated with established PF was used to explore whether patients with PF could be identified by single ILD-associated CEIs, ANN I, or ANN II. Compared with single ILD-associated CEIs, ANNs I and II had better recognition for these patients (TPR = 82.8%, Fig. 4A). Consistent with this, a dot column chart visually showed that patients with high PF risk and those that had been complicated with PF could be successfully identified with higher sensitivity and specificity using ANN II (Fig. 4B). A validation set of patients from the first affiliated hospital of HMU was used for further assessing the model effect of ANN II (Table S3, Fig. 4C). The AUC was 0.792, implying the effectiveness of ANN II to identify ILD A TPR of 84.9% was calculated for identifying ILD by drawing a 2 × 2 contingency table of the validation set (Table S4).

## DISCUSSION

Approximately 10% of patients with RA have ILD-related complications, leading to varying degrees of functional and structural impairment of the lungs (*Olson et al., 2011*; *Zou et al., 2012*; *Richman et al., 2013*). This demands clinical management targeted to those patients with RA-associated ILD, especially those with RA-associated fibrotic-ILD (*Wells & Denton, 2014*; *Mori, 2015*). For this purpose, it is important to develop early identification of patients with high risk of pulmonary complications (*Moua et al., 2014*; *Giles et al., 2014*). In the present study, a global analysis of 32 CEIs was performed to reveal clinical predictors that
related to high risk of fibrotic-ILD. A modified ANN model was built for CEI integration and detection of risk of fibrotic-ILD among patients with RA.

In this study, the Youden index, instead of Spearman's rho, was applied to explore potential associations between ILD status and CEIs without any ANN data transformation. The choice of Youden index for screening disease-associated CEIs was based on the consideration that the presence or absence of ILD belonged to a logical variable rather than a continuous variable. By calculating the Youden index, four of the 32 CEIs (age, EO, PLT, and WBC) were identified as having a relatively closer association with ILD status. This result implies multifactorial involvement of pulmonary complications in RA. Among the four ILD-associated CEIs, age showed the strongest association with ILD status (Youden index = 0.301). Older age might lead to a greater risk of fibrotic ILD. This result was in line with a previous study performed by *Yilmazer et al. (2016)*. Their study suggested that age was an independent risk factor for lung damage caused by RA-associated ILD. Obvious immunity system alterations and high infection rate were observed in RA patients with ILD (*Papanikolaou et al., 2015*; *Zamora-Legoff et al., 2016*). In the present study, EO, PLT, and WBC were found to be significantly associated with RA-ILD. White blood cells are a large category of immune system cells, which prevent damage to the body caused by foreign invaders and infectious disease Eosinophils are a type of white blood cell, while it has been traditionally recognized that platelets are a type of blood cell which plays a central role in physiological hemostasis and pathological thrombosis. Just recently, it was discovered that platelets also act as inflammatory effector cells and are importantly implicated in pathological infectious and immune responses of the lungs (*Middleton, Weyrich & Zimmerman, 2016*). Taken together, abnormalities in the count of immune cells suggest that the lungs of patients with RA-ILD might be subject to infection leading to further tissue damage, such as pulmonary fibrosis. Obviously, aging reduces the ability of the immune system to prevent disease (*Weyand & Goronzy, 2016*). However, further investigation will be necessary.

A PPSN was established by refining the four ILD-associated CEIs as coordinates in a four-dimensional space and calculating Euclidean distance between any two patients. The existence of a huge cluster of similar patients in the network implied a concentrated distribution of the majority of patients based on the results of routine medical tests (Fig. 2). Despite this, it was found that subtle differences in CEIs contributed to heterogeneous local clustering of patients with healthy lungs and those complicated with ILD in the PPSN, suggesting a possibility of joint application of the four CEIs to identify risk of PF.

ANN is a universal machine learning method nowadays, which has been widely applied in various areas of medicine, such as decision-making for neurosurgery and prostate cancer diagnosis (*Hu et al., 2013*; *Sheikhtaheri, Sadoughi & Hashemi Dehaghi, 2014*; *Azimi et al., 2015*). Application of ANN in medicine has been validated to facilitate disease risk detection and medical decision-making. In our study, a decision-making system with novel ANN model architecture was developed for the purpose of facilitating identification of high PF risk among patients with RA. Our results confirmed that integrated processing of ILD-associated CEIs by the derivative information integration system developed here more effectively identified RA patients with high risk of PF, compared with data processing the same

CEIs using a traditional ANN (Table 2). Construction of the system differed from the procedure used to build a traditional ANN, requiring multiple operations, including ANN-based CEI integration, establishment of a Euclidean distance-based PPSN, and patient derivative information extraction and integration using an ANN. Although more complex, it was thought that the system would be feasible for clinical practice, because only four routine CEIs were required as network inputs, the model architectures of ANNs I and II were simple, and the Euclidean distance calculation was easy to perform by computer programming.

In conclusion, our study for the first time investigated associations between routine CEIs and ILD in patients with RA using a modified ANN system. The results contributed to new knowledge regarding the identification of patients with high PF risk when routine CEIs were used. Integration of CEIs in a mathematical model facilitated their application in clinical management of such patients. Superior to the traditional ANN model, the developed system consisting of ANNs I and II successfully identified patients at high risk of PF and those having PF among patients with RA by co-considering meaningful associations between CEIs and ILD and similar patients in medical testing. However, two limitations should be noted: our findings were obtained from a small collection of RA patients and only 32 routine CEIs were investigated for their association with ILD. Further research on a larger patient set is definitely needed to validate our results and more CEIs should be considered for investigation.

### Funding
This work was supported by research grants from Heilongjiang Province Youth Fund (QC2013C083), Heilongjiang Province Education Bureau (12531617), Heilongjiang Province R & D project for Applied Technology (PS13H14), and National 863 Program (2006AA02Z4B1). The funders had no role in study design, data collection and analysis, decision to publish, or preparation of the manuscript.

### Grant Disclosures
The following grant information was disclosed by the authors:
Heilongjiang Province Youth Fund: QC2013C083.
Heilongjiang Province Education Bureau: 12531617.
Heilongjiang Province R & D project for Applied Technology: PS13H14.
National 863 Program: 2006AA02Z4B1.

### Competing Interests
The authors declare there are no competing interests.

### Author Contributions
- Yao Wang conceived and designed the experiments, performed the experiments, analyzed the data, contributed reagents/materials/analysis tools, wrote the paper, prepared figures and/or tables, reviewed drafts of the paper.

- Wuqi Song performed the experiments.
- Jing Wu wrote the paper.
- Zhangming Li performed the experiments, analyzed the data, prepared figures and/or tables.
- Fengyun Mu, Yang Li and He Huang contributed reagents/materials/analysis tools.
- Wenliang Zhu performed the experiments, analyzed the data, wrote the paper.
- Fengmin Zhang conceived and designed the experiments, reviewed drafts of the paper.

### Human Ethics

The following information was supplied relating to ethical approvals (i.e., approving body and any reference numbers):

The Harbin Medical University Ethical approved the study (Ethical Application Ref: HMUIRB20150028).

### Data Availability

The raw data has been supplied as a Supplementary File.

### Supplemental Information

Supplemental information for this article can be found online at http://dx.doi.org/10.7717/peerj.3021#supplemental-information.

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
