# Peer review of "Modeling using clinical examination indicators predicts interstitial lung disease among patients with rheumatoid arthritis"

_PeerJ, doi:10.7717/peerj.3021_

## Round 0.1 · original submission · Major Revisions

Both referees have picked up that the validation set of samples is not appropriate. The validation should be done on an unrelated set of samples.

Reviewer 1 ·

Basic reporting

The layout of the paper is acceptable

Experimental design

1. No information is given on how the CEIs were selected.
2. The training set contianed a mixture of RA patients with and without pulmonary fibrosis. The validation set of 87 patients all appear to have been selected bacause they had fibrosis. Usually validation sets should reflect the make up of the training set ie they should be a mixture and the ability of the test to accurately predict the presence or absence of fibrosis can then be measured.

Validity of the findings

This is difficult to assess as the validation set is biased to the presence of fibrosis ie the ability of the test to predict the abscence of fibrosis has not been looked at.

·

Basic reporting

This paper is reasonably well written. It use complex statistical and machine learning language that is somewhat difficult to follow with limited explanation of the statistical test used. For example specific values are given for the Youden index with no explanation of what they mean.
Some of the biological discussion is also very weak, for example lines 226-228.

Experimental design

The study is a simple prospective clinical study.
There is a limited explanation of the ethical approvals used for the study.
It is not clear if the training set and the validation set are appropriate. It appears that the "validation" was performed by using a set of patients with known RA associated ILD. Perhaps it would have been better had the authors used a second unrelated cohort of patients with the same ILD distribution as the training set?

This is a major flaw as it will lead to a significant bias.

Validity of the findings

I am not sure that the data is very robust for the reasons outlined above.
The biology of the findings are also not very clear.

Additional comments

The authors set out to build a novel statistical model that they propose will allow the identification of RA patients at risk of developing ILD and pulmonary fibrosis. ILD is a recognized complication of RA and the rational for the study is reasonably well made.

The model uses a large number of routinely collected clinical examination indicators and the analysis suggests that a number of these might have some utility in defining RA patients at risk of developing ILD. To validate the model they use a test cohort of patients with the diagnosis of RA associated ILD. Unsurprisingly, this provides support for the model.

I am not sure that this is an appropriate validation set. Perhaps it would have been better had the authors used a second unrelated cohort of patients with the same ILD distribution as the training set? They appear to recognise this as a weakness in the discussion.

I felt that the paper was written in a way that made it rather difficult to understand the underlying biology. Much is made about the data but there is very little discussion about why specific CEIs might be associated with ILD development. It is unclear if the authors are suggesting the use of the 4 CEIs alone will have utility in identifying patients at risk of developing RA associated ILD. As such I am unclear as to the value of the paper as written.

---

## Round 0.2 · accepted · Accept

The manuscript has been much improved by the addition of a new test set of data. The statistics appear to be correctly applied.